# Self-Adjusting Ant Colony Optimization Based on Information Entropy for Detecting Epistatic Interactions

**DOI:** 10.3390/genes10020114

**Published:** 2019-02-01

**Authors:** Boxin Guan, Yuhai Zhao

**Affiliations:** Key Laboratory of Intelligent Computing in Medical Image, Ministry of Education, and School of Computer Science and Engineering, Northeastern University, Shenyang 110819, China; neuguanboxin@163.com

**Keywords:** single nucleotide polymorphisms, ant colony optimization, information entropy, epistatic interactions, self-adjusting algorithm

## Abstract

The epistatic interactions of single nucleotide polymorphisms (SNPs) are considered to be an important factor in determining the susceptibility of individuals to complex diseases. Although many methods have been proposed to detect such interactions, the development of detection algorithm is still ongoing due to the computational burden in large-scale association studies. In this paper, to deal with the intensive computing problem of detecting epistatic interactions in large-scale datasets, a self-adjusting ant colony optimization based on information entropy (IEACO) is proposed. The algorithm can automatically self-adjust the path selection strategy according to the real-time information entropy. The performance of IEACO is compared with that of ant colony optimization (ACO), AntEpiSeeker, AntMiner, and epiACO on a set of simulated datasets and a real genome-wide dataset. The results of extensive experiments show that the proposed method is superior to the other methods.

## 1. Introduction

A single nucleotide polymorphism (SNP) is a variation in a single nucleotide that occurs at a specific position in the genome. SNPs are widespread in the genome and easy to detect, so they are typically used as biomarkers in genome-wide association studies (GWAS) [1,2,3]. Early GWAS focused on the association between single-locus and phenotypes, and they have achieved a great deal in the research of single-gene diseases [4,5]. For many single-gene diseases, also known as Mendelian diseases, researchers have been able to find the corresponding pathogenic genes [6,7]. Nevertheless, owing to the sophisticated regulation mechanism in the human genome, the genetic basis of numerous complex diseases is still unknown [8,9,10,11]. It is universally acknowledged that these sophisticated traits are due to the combined action of multiple genetic variations rather than of a single variation [12,13]. These multiple genetic variations may demonstrate very minor influences alone but jointly they have strong influences; this is known as multi-locus or epistatic interaction [14,15].

Current approaches for detecting epistatic interactions can generally be classified into four categories: exhaustive search methods [16,17,18,19,20], stochastic search methods [21,22,23,24], evolutionary computing methods [25,26,27,28,29], and machine learning methods [30]. Among these, exhaustive search methods can find all possible epistatic interactions but the computational burden is obvious when datasets become large. Some examples are multifactor dimensionality reduction (MDR) [16,17,18], boolean operation-based screening and testing (BOOST) [19], and tree-based epistasis association mapping (TEAM) [20]. Random methods detect epistatic interactions by random sampling, which can greatly speed up the process. Epistatic module detection (epiMODE) [21], detection of epistatic interactions using random forest (epiForest) [22], Bayesian Epistasis Association Mapping (BEAM) [23], and SNPHarvester [24] are examples. Evolutionary computational methods are stochastic algorithms that simulate biological evolutionary processes. Some evolutionary computing methods based on ant colony optimization (ACO) [31] have been proposed to detect the epistatic interactions in large-scale association studies, such as AntEpiSeeker [25], AntMiner [26], and epiACO [27]. With the development of machine learning technologies, machine learning methods are also used to detect epistatic interactions. Bayesian network (BN) [30] is one example.

The detection of epistatic interactions faces severe computational challenges. A large dataset for GWAS may have hundreds of thousands to millions of SNPs. For example, there are at least 5 billion combinations that need to be evaluated by an exhaustive search for two-locus interactions on the age-related macular degeneration (AMD) dataset [32]. Stochastic search methods and evolutionary computing methods have been shown to be able to handle large-scale datasets. However, developing a more efficient and reliable search algorithm is still desired. It has been proven that the ACO is an effective approach to solve epistatic interaction problems. However, the path selection of ants is single and blind, making the search fall into the local optimal state. Targeted at the deficiencies of the ACO, a modified ACO, self-adjusting ant colony optimization based on information entropy (IEACO), is presented to solve large-scale epistatic interaction problems. Through automatically self-adjusting the path selection strategy, IEACO is capable of maintaining the diversity of solutions, and accordingly, improve the quality of the solutions. In order to evaluate the detection power of this method, we conducted an experiment on a cluster of simulated datasets and a real whole-genome dataset and did a comparative study of the performance of ACO, AntEpiSeeker, AntMiner, epiACO, and IEACO. As suggested by the computer simulation results, IEACO has an edge over all the other approaches. In addition, the feasibility of IEACO was also verified by a real whole-genome experiment. 

## 2. Materials and Methods 

### 2.1. Problem Definition

This paper primarily concentrates on case-control research based on the hypothesis that overall SNPs are biallelic. With known genotype data at *L* SNPs of *S* samples, we used *S_1_* and *S_2_* to express the number of controls and the number of cases, respectively. *r_k_* is used to indicate the kth SNP (1 ≤ *k* ≤ *L*). *Y* is used to indicate the state of disease, where 1 stands for case and 2 stands for control. We used capital letters (A, B) to represent major alleles and lowercase letters (a, b) to represent minor alleles. In accordance with the copy number of minor alleles at each locus, the genotype can be set to either 0, 1, or 2. 

### 2.2. Standard Ant Colony Optimization

Ant colony optimization (ACO) is a successful approach that is useful in the solution of NP-hard combination optimization problems and it has been widely applied in GWAS. The basic idea of ACO is to express the feasible solutions of optimization problems with ant paths and use overall paths of the ant group to constitute the solution space of optimization problems. Ants on relatively short paths tend to release more pheromone. With the passage of time, pheromone concentration that accumulates on the short paths gradually increases and more and more ants choose the paths. Eventually, all the ants will gather on the optimal path under positive feedback, which exactly corresponds to the optimal solution of the optimization problem. 

Artificial ants choose loci per the following formula to solve epistatic interaction problems:(1)$$p_{k}(i)=\frac{{\left(\tau_{k}(i)\right)}^{\alpha }\eta_{k}^{\beta }}{\sum_{j=1}^{L}{\left(\tau_{j}(i)\right)}^{\alpha }\eta_{j}^{\beta }}$$
where *τ_k_*(*i*) denotes the pheromone value on locus *k* at iteration *i*; *η_k_* denotes the heuristic factor on locus *k*; *α* denotes the pheromone weight; and *β* denotes the heuristic weight. The value of pheromone can be upgraded by the following formula: (2)$$\tau_{k}(i+1)=(1-\rho )\tau_{k}(i)+\Delta \tau_{k}(i)$$
where *ρ* denotes the pheromone evaporation rate; ∆*τ_k_*(*i*) denotes the variation of pheromone value (∆*τ_k_*(*i*) = χ^2^); and χ^2^-test is the fitness function. In terms of epistasis detection problems, the null hypothesis is that there is no epistatic interaction in the dataset. The alternative hypothesis is that the *p*-values in one or multiple epistatic interactions are below the significance level.

The search procedure for solving the epistatic interaction problem is as follow. First, ACO initializes *M* ants and sets up an identical pheromone value for each locus. Second, each ant randomly chooses an SNP set with *K* locus. Third, we assess each chosen SNP set through the χ^2^-test and upgrade the pheromone of each locus. Then, we determine the set with the highest χ^2^-value as the candidate and calculate the corresponding *p*-value. The optimization process should be repeatedly conducted until the number of iterations totals the preset value. In the end, the candidates whose *p* values are smaller than a Bonferroni-corrected significance threshold are reported.

However, when applying ACO to solve epistatic interactions, it can make the ants select an excessively simple path, easily falling into local optimum. Moreover, owing to its inability to provide real-time information feedback to regulate the behavior of the ants, ACO usually does not guide the search direction. 

### 2.3. Self-Adjusting Ant Colony Optimization Based on Information Entropy

To improve the efficiency of ACO, information entropy is incorporated and the novel algorithm is called self-adjusting ant colony optimization based on information entropy. Entropy is a measure of the uncertainty of information. The greater the entropy, the higher the uncertainty of the information, as defined by Equation (3). In the equation, *p* is the probability of a particular event occurring, and the logarithm takes 2 as the base. For ACO, *p_k_*(*i*) is the proportion of pheromone on locus *k* to total pheromones at iteration *i*, and it is greater than or equal to 0. The information entropy is the biggest when each locus has an equal pheromone value. With the convergence of the algorithm, the amount of pheromone is concentrated on some loci, and the information entropy is reduced.
(3)$$H(i)\;=-\sum_{k=1}^{L}p_{k}(i)\log p_{k}(i)$$

Based on the information entropy, IEACO can automatically adjust the path selection strategy according to Equation (4). This equation gives 2 selection strategies: the first is the path selection strategy of standard ACO, which is called the positive feedback strategy; the other is the improved path selection strategy, which is called the negative feedback strategy. In the equation, *H*(*i*), *w_k_*(*i*), and *γ* are newly added. *H*(*i*) is the information entropy at iteration *i*; *w_k_*(*i*), defined by Equation (5), is the negative feedback pheromones for locus *k* at iteration *i*; and *γ* is a parameter determining the weight of negative feedback pheromones. In Equation (5), *μ* can be seen as the upper bound of negative feedback pheromones on the worse loci. With deepening of the iteration, the pheromone values of *μ*–*τ_k_* (*i*) on the bad loci are increased, thereby increasing the probability that these loci are selected. |*H*(*i*)–*H*(*i*–1)| represents the difference between the information entropy of 2 adjacent iterations. When the difference is not greater than a specified switch parameter *θ*, IEACO uses the newly added negative feedback strategy to select loci. This small change in the information entropy indicates that the algorithm has been converging, so it is obvious that utilizing this new equation can increase the probability that artificial ants choose bad loci.
(4)$$p_{k}(i)=\;\left\{ \begin{array}{ll} \frac{{\left(\tau_{k}(i)\right)}^{\alpha }\eta_{k}^{\beta }}{\sum_{j=1}^{L}{\left(\tau_{j}(i)\right)}^{\alpha }\eta_{j}^{\beta }} & \left|H(i)-H(i-1)\right|>\theta \\ \frac{{\left(\tau_{k}(i)\right)}^{\alpha }{\left(\eta_{k}(i)\right)}^{\beta }{\left(w_{k}(i)\right)}^{\gamma }}{\sum_{j=1}^{L}{\left(\tau_{j}(i)\right)}^{\alpha }{\left(\eta_{j}(i)\right)}^{\beta }{\left(w_{j}(i)\right)}^{\gamma }} & otherwise \end{array}\right.$$
(5)$$w_{k}(i)=\mu -\tau_{k}(i)$$

Self-adjusting ant colony optimization based on information entropy searches for epistatic interactions by constantly updating the information entropy and self-adjustment of the ant behavior. At the beginning of the algorithm, the pheromones on each path are equal and the information entropy is the biggest. As the number of iterations increases, the pheromones on the good paths increase while the pheromones on the bad paths decrease. At the same time, the information entropy continues to decrease. When the difference between the information entropy of 2 adjacent iterations is very small, the algorithm changes the strategy of how ants choose paths. IEACO takes advantage of history searching information and dynamically guides ant swarms to explore unknown space during the optimization procedure.

The pseudo-code of IEACO to solve the epistatic interactions is given in Figure 1. IEACO calculates the information entropy *H*(*i*) at current iteration *i*. After each ant finds an SNP combination, each SNP combination is evaluated by the *χ*^2^-test, and the SNP combination with the highest *χ*^2^-value is recorded. Then, a path selection strategy is used according to a comparison of the difference between the information entropy of 2 adjacent iterations and the specified switch parameter *θ*.

### 2.4. An Example of Self-Adjusting Ant Colony Optimization Based on Information

There are 10 SNPs (r1, r2, r3, r4, r5, r6, r7, r8, r9, and r10), and their initial pheromones are 100. In this example, suppose that the interaction is embedded in the position of (r2, r9). ACO and IEACO respectively generate 3 artificial ants to search through 3 iterations in this space. Figure 2a gives the search process of ACO. *k* is the SNP locus, *τ_k_*(*i*) is the pheromone of locus *k* at iteration *i*, and *p_k_*(*i*) is the proportion of pheromones on locus *k* to total pheromones at iteration *i*. At iteration 1, each ant selects an SNP set ((r1, r2), (r3, r4), and (r5, r6)), and the χ^2^-value for each SNP set is calculated. It is assumed here that the χ^2^-values between different loci are known (χ^2^(r1, r2) = 50, χ^2^(r3, r4) = 40, and χ^2^(r5, r6) = 20). Then, the SNP set with the highest χ^2^-value is recorded as a candidate solution (χ^2^(r1, r2) = 50), and the pheromone for each locus is updated. The above procedure is executed for 3 iterations. Finally, the 3 recorded candidate solutions are compared and the interaction (r1, r2) with the highest χ^2^-value is output. If the interaction position detected by the algorithm is inconsistent with the real position, the result is a false positive. The search process of IEACO is shown in Figure 2b. Compared with ACO, IEACO calculates the information entropy before each iteration to select the search strategy. The positive feedback strategy is the default at iteration 1. Here, the upper bound of negative feedback pheromone on worse paths *μ* is 300, and the switch parameter *θ* is set to 0.01. For iteration 2, the positive feedback strategy is selected again due to |*H*(1)−*H*(0)| = 0.03 > *θ*. After completing this iteration, |*H*(2)−*H*(1)| = 0 ≤ *θ*, so the negative feedback strategy is adopted. Loci 9 and 10, with the least amount of pheromones, are selected as negative feedback loci. The pheromones of the 2 loci are recalculated (*w*_9_(2) = *μ–τ*_9_(2) = 219 and *w*_10_(2) = *μ–τ*_10_(2) = 219), and their corresponding probabilities are increased. Finally, the correct solution (r2, r9) is obtained through iteration 3.

## 3. Results and Discussion

### 3.1. Data Preparation and Parameter Setting

We compared the proposed IEACO algorithm with ACO, AntEpiSeeker [25], AntMiner [26], and epiACO [27] on a wide range of simulated datasets. The reason for choosing these algorithms is that they are ACO-based methods and have shown their detection power in a variety of disease models. The datasets contain eight commonly used two-locus epistasis models [33,34], and the details of these models are shown in Table 1. Models 1 and 2 are multiplicative models with marginal effects, and Models 3 and 4 are threshold models with marginal effects [35]. The other four are epistasis models without marginal effects. Models 5 and 6 are directly cited from [33], Model 7 is a ZZ model [36], and Model 8 is an XOR model [33]. For each model, 200 datasets were generated [37], and each dataset contains 2000 samples (1000 cases and 1000 controls). There are 500 SNPs in the first 100 datasets, and the number of SNPs increases to 5000 in the other 100 datasets. Minor allele frequency (MAF) refers to the frequency at which the second most common allele occurs in a given population. Prevalence is the proportion of a particular population found to be affected by a disease.

It is essential to ensure that the computational effort of different comparative methods is set up equally. In the five ACOs, the number of calls of the fitness function is equal to the number of ants in an iteration. Therefore, the same number of iterations makes the fitness function the same for each algorithm. For the two-locus epistasis detecting experiment, the number of iterations is *N* = 0.2 × number of SNPs, the number of ants *M* is 500 on 500-SNP datasets and 200 on 5000-SNP datasets, the initial pheromone *τ* is 100, the heuristic factor *η* is 1, the weight parameters α and *β* are 1, and the evaporation rate *ρ* is 0.05. For AntEpiSeeker, AntMiner, and epiACO, the other parameter settings are set as default values [25,26,27]. For IEACO, the upper bound of negative feedback pheromone on worse paths *μ* is 300, the parameter determining the weight of negative feedback pheromone *γ* is 1, the switch parameter *θ* is 0.001. These parameters of IEACO were obtained by running this algorithm many times on the same dataset.

### 3.2. Detection Power Comparison

The detection power is the ratio of the number of successful identifications to the number of all experimental datasets. The proposed IEACO algorithm was compared with ACO, AntEpiSeeker, AntMiner, and epiACO on 500-SNP and 5000-SNP datasets, and the experimental results are shown in Figure 3 and Figure 4. The horizontal axis represents the eight models, and the vertical axis represents the percentage of interactions correctly identified by each tested algorithm. As seen from the figures, IEACO is not significantly superior to the other algorithms on 500-SNP datasets, but the method still has the best detection power because it identifies all the epistatic interactions on four models. The detection power on 5000-SNP datasets is lower than that on 500-SNP datasets because of the increased search space. The detection power of IEACO is significantly superior to that of the other tested algorithms on 5000-SNP datasets. This is because IEACO automatically adjusts the behavior of ants to increase the possibility of searching for interactions in a large-scale search space. In addition, Table 2 shows the mean and standard deviation of IEACO. For each model, all of the power results solved by the proposed algorithm in 30 runs are used as experimental data. As can be seen from the figures, the resulting distribution of IEACO is concentrated. Thus, it can be concluded that the proposed algorithm is stable and powerful.

### 3.3. Recall and Precision Analysis

Recall and precision, concepts in the field of information retrieval, are important indicators that reflect the search effect. Recall is the number of resulting true positives divided by the total number of true positives in all datasets as in Equation (6), while precision is the number of resulting true positives divided by the sum of resulting values as in Equation (7).
(6)$$\mathrm{Recall}\;=\;\frac{\mathrm{True}\;\mathrm{Positive}\;(\mathrm{TP})}{\mathrm{True}\;\mathrm{Positive}\;(\mathrm{TP})\;+\;\mathrm{False}\;\mathrm{Negative}\;(\mathrm{FN})\;}$$
(7)$$\mathrm{Precision}=\frac{\mathrm{True}\;\mathrm{Positive}\;(\mathrm{TP})}{\mathrm{True}\;\mathrm{Positive}\;(\mathrm{TP})\;+\;\mathrm{False}\;\mathrm{Positive}\;(\mathrm{FP})}$$

The results of recall and precision were recorded and are shown in Table 3 and Table 4. The recall is represented as R, and the precision is represented as P in the two tables. The algorithm with the highest recall or accuracy on a model is displayed in bold font. For 500-SNP datasets, IEACO has the highest recall in five of the eight models, and it achieved the highest precision in six of them. AntMiner and epiACO also achieved high recall and precision. For 5000-SNP datasets, the recall and precision of IEACO are the highest of all eight models, and these which are obviously superior to those of the other algorithms. In conclusion, most of the results indicate that the proposed IEACO algorithm is an effective method for detecting epistatic interactions.

### 3.4. Hypothesis Test

To prove the validity of IEACO in solving the epistatic interaction problem, hypothesis tests were used. Hypotheses included Holm (Holm), Hochberg (Hoch), and Hommel (Homm) [38,39]. The power values of the eight models were combined as experimental data. The comparative analysis results for IEACO with ACO, AntEpiSeeker, AntMiner, and epiACO are shown in Table 5 and Table 6. These procedures reject the hypotheses when *p*-values ≤ 0.05. On 500-SNP datasets, IEACO is not significantly superior to AntEpiSeeker, AntMiner, and epiACO. On 5000-SNP datasets, IEACO is significantly superior to the other algorithms on all hypothetical tests. This shows that the proposed algorithm has a natural advantage in solving the epistatic interaction for large-scale data.

### 3.5. Execution Time Analysis

Since execution time is also a critical indicator for evaluating the performance of detection algorithms, we conducted a comparative study of the time performance of five algorithms on the eight models. The average running times with their respective standard deviations were recorded in terms of seconds per run, and the data are shown in Table 7 and Table 8. ACO is the fastest and IEACO has the second fastest execution speed, because it consumes more time with its self-adjusting strategy. The next two algorithms are AntEpiSeeker and epiACO, because the post-processing of AntEpiSeeker is time-consuming and the two evaluation objective calculations of epiACO. Because it uses some prior knowledge, AntMiner has the slowest execution time. IEACO, which performed best in the previous experiments, does not have an advantage in time performance; in comparison to the fastest ACO, the execution time of IEACO does not greatly increase. Based on the above analysis, we can conclude that IEACO will not improve performance at the cost of increasing the execution time. 

### 3.6. Power Comparison of Multiple Epistasis Detection

To prove the power of IEACO for detecting multiple epistatic interactions, we compared the method with ACO, AntEpiSeeker, AntMiner, and epiACO on eight 1000-SNP simulated datasets (Case 1–Case 8). Each of the eight cases is composed of five pure epistatic models [40] with the same MAF and heritability. The top five two-SNP interactions with *p*-values below significance are reported in Figure 5. The detection power is divided into three parts: the percentage of true positive epistatic interactions, the percentage of false positive epistatic interactions, and the percentage of undetected epistatic interactions. The detection ability of ACO is low for true positives and high for false positives. Although the number of true positives identified by AntEpiSeeker is small in some cases, the number of true positives identified by the algorithm is also small. Although AntMiner and epiACO perform well in detecting true positives, they have poor detection performance for false positives. Compared with the other algorithms, IEACO identifies the largest percentage of true positive epistatic interactions and the lowest percentage of false positive epistatic interactions on most large-scale datasets, which suggests that IEACO significantly outperforms the other algorithms. 

### 3.7. Results from Age-Related Macular Degeneration Dataset

The feasibility of IEACO was verified by analyzing a real age-related macular degeneration (AMD) dataset containing 146 samples (96 cases and 50 controls) and 116,204 SNPs [32]. Table 9 presents the epistatic interactions related to AMD. It follows that most epistatic interactions contain either rs380390 or rs1329428, because these two SNPs have the strongest principal effects among overall SNPs. As a result, the combination of the two with other SNPS usually demonstrates relatively strong interaction effects and identification prominence. SNPs rs380390 and rs1329428 exist in the gene CFH, where the mutation of the gene has a close connection with hemolyticuremic syndrome and chronic hypocomplementemic nephropathy. The two loci are also thought to be significantly correlated with AMD [21]. SNP rs1394608, which is located in the SGCD gene of chromosome 5, has also been reported to be strongly associated with AMD [21], and it is speculated that the interaction of rs1394608 with rs1740752 influences AMD. However, IEACO merely detects epistatic interactions, excluding rs380390 and rs1329428. The possible correlation between AMD and the interactions (rs994542, rs9298846) and (rs1740752, rs1368863) is detected. The four SNPs in the noncoding region perhaps result in AMD by regulating gene expression level. These results verify the feasibility of the proposed IEACO algorithm and may provide some clues for exploring the pathogenic factors of AMD.

## 4. Conclusions

The detection of epistatic interactions is one of the most critical stages in GWAS application. Actually, epistatic interactions not only play a vital role in the research on the pathogenesis of complex diseases triggered by geno-variation, but are also highly valuable to real research of SNPs. As a result, it is essential to reinforce the accuracy of detection algorithms and reduce their time complexity. In this paper, the proposed IEACO algorithm is a simple but effective method for solving epistatic interactions. It can automatically self-adjust the path selection strategy of ants as per real-time information, and accordingly, it has better epistatic interaction detection capacity. Simulated and AMD datasets were used to evaluate the performance of IEACO. As proved by the experimental results, the algorithm has significant efficiency and feasibility. 

## Figures and Tables

**Figure 1 genes-10-00114-f001:**
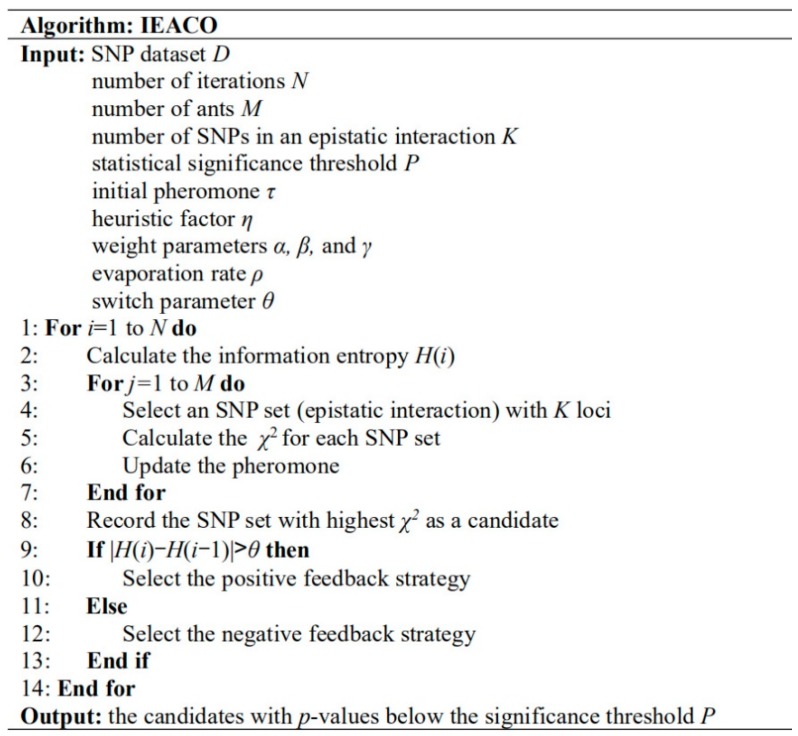
Pseudocode of ant colony optimization based on information entropy (IEACO) to solve epistatic interactions.

**Figure 2 genes-10-00114-f002:**
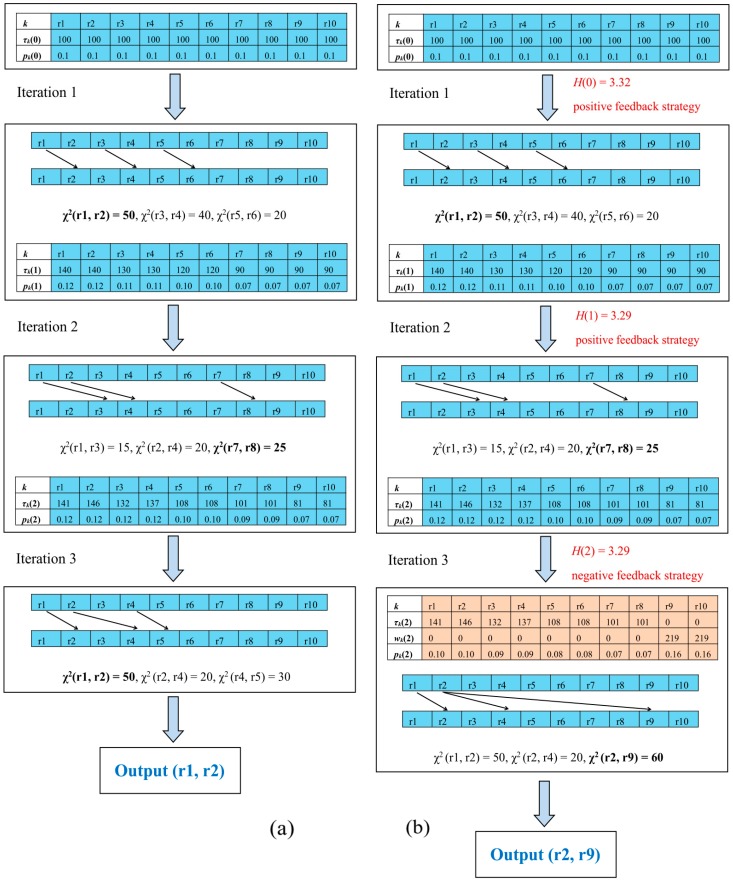
(**a**) The process of searching for interactions by using ant colony optimization (ACO); (**b**) the process of searching for interactions using IEACO.

**Figure 3 genes-10-00114-f003:**
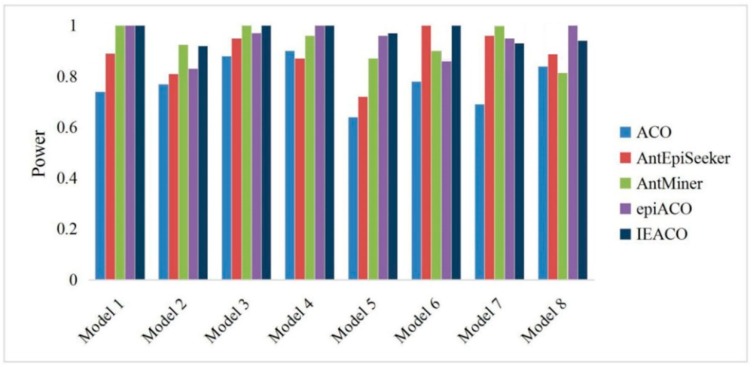
Power comparison on 500-SNP datasets.

**Figure 4 genes-10-00114-f004:**
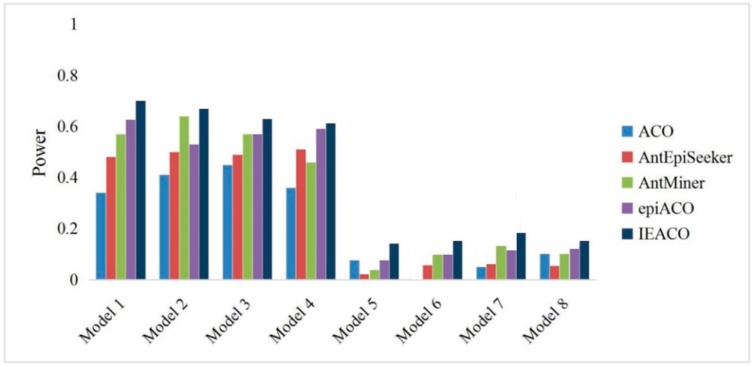
Power comparison on 5000-SNP datasets.

**Figure 5 genes-10-00114-f005:**
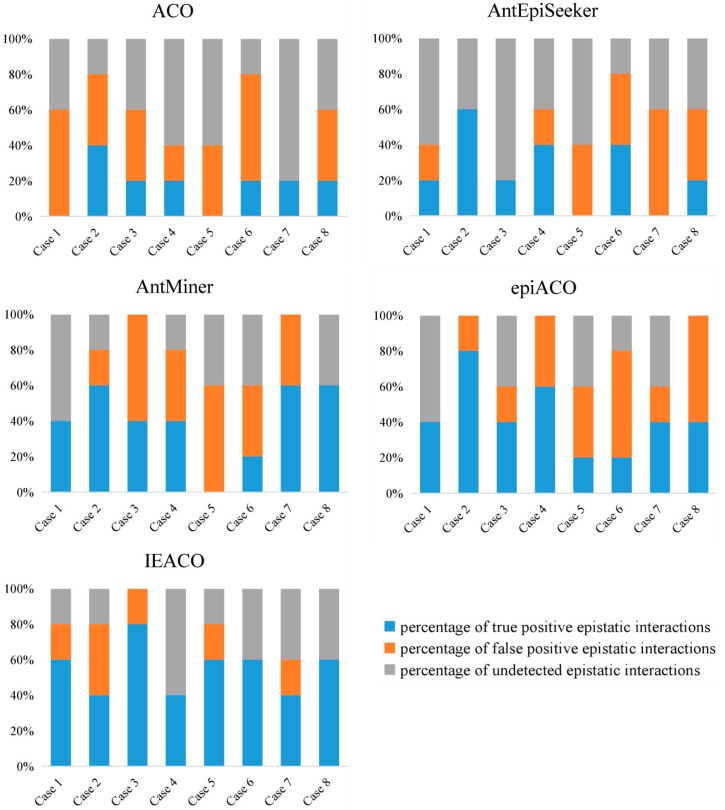
Performance comparison of multiple epistasis detection.

**Table 1 genes-10-00114-t001:** Details of eight commonly used two-locus epistasis models.

Model 1	Prevalence = 0.100, MAF(a) = 0.30, MAF(b) = 0.20			Model 2	Prevalence = 0.100, MAF(a) = 0.20, MAF(b) = 0.20		
	AA	Aa	aa		AA	Aa	aa
BB	0.087	0.087	0.087	BB	0.092	0.092	0.092
Bb	0.087	0.146	0.190	Bb	0.092	0.145	0.181
bb	0.087	0.190	0.247	bb	0.092	0.181	0.227
Model 3	Prevalence = 0.100, MAF(a) = 0.05, MAF(b) = 0.05			Model 4	Prevalence = 0.100, MAF(a) = 0.50, MAF(b) = 0.50		
	AA	Aa	aa		AA	Aa	aa
BB	0.096	0.096	0.096	BB	0.052	0.052	0.052
Bb	0.096	0.533	0.533	Bb	0.052	0.137	0.137
bb	0.096	0.533	0.533	bb	0.052	0.137	0.137
Model 5	Prevalence = 0.064, MAF(a) = 0.20, MAF(b) = 0.20			Model 6	Prevalence = 0.171, MAF(a) = 0.40, MAF(b) = 0.40		
	AA	Aa	aa		AA	Aa	aa
BB	0.486	0.960	0.538	BB	0.068	0.299	0.017
Bb	0.947	0.004	0.811	Bb	0.289	0.044	0.285
bb	0.640	0.606	0.909	bb	0.048	0.262	0.174
Model 7	Prevalence = 0.038, MAF(a) = 0.50, MAF(b) = 0.50			Model 8	Prevalence = 0.010, MAF(a) = 0.50, MAF(b) = 0.50		
	AA	Aa	aa		AA	Aa	aa
BB	0.000	0.000	0.100	BB	0.000	0.020	0.000
Bb	0.000	0.050	0.000	Bb	0.020	0.000	0.020
bb	0.100	0.000	0.000	bb	0.000	0.020	0.000

**Table 2 genes-10-00114-t002:** Mean power of IEACO with its respective standard deviation.

	Model 1	Model 2	Model 3	Model 4	Model 5	Model 6	Model 7	Model 8
500-SNP	100 ± 0.00	92 ± 1.85	100 ± 0.00	100 ± 0.00	97 ± 1.06	100 ± 0.00	93 ± 2.59	94 ± 2.08
5000-SNP	70 ± 4.03	68 ± 4.20	63 ± 2.89	61± 4.56	16 ± 3.03	17 ± 3.14	20 ± 1.99	17 ± 2.05

**Table 3 genes-10-00114-t003:** Results of recall and precision on 500-SNP datasets.

	ACO		AntEpiSeeker		AntMiner		epiACO		IEACO	
	R	P	R	P	R	P	R	P	R	P
Model 1	0.74	0.74	0.89	0.76	**1**	0.78	**1**	0.82	**1**	**0.83**
Model 2	0.77	0.83	0.81	0.84	**0.93**	0.92	0.83	0.89	0.92	**0.93**
Model 3	0.88	0.78	0.95	0.88	**1**	0.89	0.97	0.79	**1**	**0.90**
Model 4	0.90	0.89	0.87	0.91	0.96	0.82	**1**	0.84	**1**	**0.92**
Model 5	0.64	0.72	0.72	0.74	0.81	**0.88**	0.96	0.80	**0.97**	0.85
Model 6	0.78	0.82	**1**	0.86	0.90	0.93	0.86	0.91	**1**	**0.94**
Model 7	0.69	0.65	0.96	0.78	**1**	0.81	0.95	0.77	0.93	**0.84**
Model 8	0.84	0.91	0.88	0.85	0.82	0.84	**1**	**0.93**	0.94	0.89

**Table 4 genes-10-00114-t004:** Results of recall and precision on 5000-SNP datasets.

	ACO		AntEpiSeeker		AntMiner		epiACO		IEACO	
	R	P	R	P	R	P	R	P	R	P
Model 1	0.34	0.54	0.48	0.61	0.57	0.70	0.62	0.72	**0.70**	**0.74**
Model 2	0.41	0.52	0.50	0.64	0.64	0.66	0.53	0.71	**0.68**	**0.72**
Model 3	0.45	0.57	0.49	0.68	0.57	0.63	0.57	0.65	**0.63**	**0.69**
Model 4	0.36	0.48	0.51	0.55	0.46	0.59	0.59	0.64	**0.61**	**0.67**
Model 5	0.07	0.19	0.02	0.13	0.04	0.16	0.07	0.17	**0.16**	**0.24**
Model 6	0.00	0.00	0.05	0.17	0.11	0.38	0.11	0.27	**0.17**	**0.31**
Model 7	0.05	0.45	0.06	0.40	0.15	0.48	0.13	0.55	**0.20**	**0.56**
Model 8	0.12	0.63	0.06	0.56	0.12	0.44	0.14	0.50	**0.17**	**0.58**

**Table 5 genes-10-00114-t005:** Adjusted *p*-values on 500-SNP datasets.

Hypothesis	Unadjusted *p*	*p_Hoch_*	*p_Hoch_*	*p_Hoom_*
IEACO vs ACO	3.743122471085636E-4	0.00149724898843425	0.00149724898843425	0.00149724898843425
IEACO vs AntEpiSeeker	0.0577795711235972	0.17333871337079185	0.17333871337079185	0.17333871337079185
IEACO vs AntMiner	0.5270892568655381	1.0541785137310762	0.5270892568655381	0.5270892568655381
IEACO vs epiACO	0.5270892568655381	1.0541785137310762	0.5270892568655381	0.5270892568655381

**Table 6 genes-10-00114-t006:** Adjusted *p*-values on 5000-SNP datasets.

Hypothesis	Unadjusted *p*	*p_Hoch_*	*p_Hoch_*	*p_Hoch_*
IEACO vs ACO	9.546919845278683E-6	3.818767938111473E-5	3.818767938111473E-5	3.818767938111473E-5
IEACO vs AntEpiSeeker	1.478023103344183E-4	4.434069310032551E-4	4.434069310032551E-4	4.434069310032551E-4
IEACO vs AntMiner	0.01770606580736659	0.03541213161473319	0.03541213161473319	0.03541213161473319
IEACO vs epiACO	0.03983261924474151	0.03983261924474151	0.03983261924474151	0.03983261924474151

**Table 7 genes-10-00114-t007:** Results of running time on 500-SNP datasets.

	Model 1	Model 2	Model 3	Model 4	Model 5	Model 6	Model 7	Model 8
ACO	12.3 ± 0.4	14.1 ± 0.3	19.8 ± 4	10.3 ± 0.5	18.0 ± 0.4	19.1 ± 0.5	17.7 ± 0.5	11.7 ± 0.3
AntEpiSeeker	28.2 ± 0.5	27.3 ± 0.5	28.0 ± 0.6	30.3 ± 0.7	29.4 ± 0.6	36.8 ± 0.8	38.2 ± 0.8	34.5 ± 0.7
AntMiner	108.9 ± 4.1	123.4 ± 3.3	98.9 ± 3.4	100.6 ± 3.8	112.3 ± 4.4	109.9 ± 3.1	131.2 ± 4.6	132.0 ± 3.9
epiACO	25.6 ± 0.8	24.9 ± 0.5	22.2 ± 0.5	23.1 ± 0.6	20.4 ± 0.7	29.0 ± 0.7	28.8 ± 0.7	23.2 ± 0.6
IEACO	17.7 ± 0.6	18.1 ± 0.4	16.5 ± 0.4	16.8 ± 0.5	21.2 ± 0.5	22.5 ± 0.5	23.1 ± 0.6	17.0 ± 0.6

**Table 8 genes-10-00114-t008:** Results of running time on 5000-SNP datasets.

	Model 1	Model 2	Model 3	Model 4	Model 5	Model 6	Model 7	Model 8
ACO	84.7 ± 4.5	82.3 ± 5.1	78.9 ± 3.4	80.5 ± 4.2	83.4 ± 3.9	85.1 ± 4.5	79.3 ± 3.6	80.1 ± 4.1
AntEpiSeeker	187.8 ± 8.9	182.3 ± 9.1	178.9 ± 7.8	179.9 ± 8.1	186.7 ± 8.4	179.0 ± 6.9	170.3 ± 6.3	177.4 ± 7.2
AntMiner	824.1 ± 67.4	865.2 ± 76.8	778.4 ± 57.3	894.3 ± 62.6	842.2± 63.1	811.5 ± 60.7	870.1 ± 79.2	888.3 ± 80.3
epiACO	126.5 ± 74	119.7 ± 8.1	116.7 ± 7.0	115.0 ± 6.8	133.6 ± 5.7	130.2 ± 6.2	123.1 ± 5.4	120.4 ± 6.0
IEACO	93.8 ± 5.2	91.4 ± 4.2	90.3 ± 3.7	94.1 ± 4.1	97.9 ± 5.1	96.5 ± 5.0	89.4 ± 4.4	92.3 ± 3.9

**Table 9 genes-10-00114-t009:** Experimental results of age-related macular degeneration (AMD) data identified by IEACO.

SNP 1	Chromosome	Gene	SNP 2	Chromosome	Gene
rs380390	1	CHF	rs1363688	5	N/A
rs380390	1	CHF	rs2402053	7	N/A
rs380390	1	CHF	rs2224762	9	KDM4C
rs1329428	1	CHF	rs2113379	2	ADAM23
rs1329428	1	CHF	rs3922799	2	N/A
rs1329428	1	CHF	rs1822657	21	NCAM2
rs1394608	5	SGCD	rs1740752	10	N/A
rs994542	6	N/A	rs9298846	9	N/A
rs1740752	10	N/A	rs1368863	11	N/A

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
