# Peer review of "Self-Adjusting Ant Colony Optimization Based on Information Entropy for Detecting Epistatic Interactions"

_genes, 2019, doi:10.3390/genes10020114_

Round 1

Reviewer 1 Report

This manuscript is interesting and addresses a highly difficult issue. Though the method proposed is simple, it seems to improve over other methods, in a number of simulated conditions. This manuscript is the fruit of an intensive work (comparison with four other ACO-based methods, on 8 models). Unfortunately, the quality of English definitely prevents acceptance of the article unless extensive editing is performed by a native English speaker.

We encourage the authors to revise their manuscript, write a cover letter with a reminder of each current defective paragraph and close below, the novel modified paragraph, for an easy check by the reviewers.

Author Response

Response to Reviewer 1 Comments

Point 1: We encourage the authors to revise their manuscript, write a cover letter with a reminder of each current defective paragraph and close below, the novel modified paragraph, for an easy check by the reviewers.

Response 1: Thank you for reading our manuscript carefully. We have revised our manuscript by the MDPI English editing service. A manuscript of the grammar correction has been submitted.

Reviewer 2 Report

The manuscript handles an interesting topic in bioinformatics.  In general, the quality of the paper should be improved, and I have some concerns regarding the originality and contribution of this work, as detailed below. The authors should consider the following comments before accepting the manuscript for publication in this prestige journal.

A. The comparison of methods is questionable. I recommend you improve this part of the paper (for example using ideas from the following article). References: Garcia, S., Herrera, F. (2008). An extension on statistical comparisons of classifiers over multiple data sets for all pairwise comparisons. Journal of Machine Learning Research 9:2677-2694.

B. Are the achieved results statistically significant? (between the methods)

C. The authors define N as the number of iteration of the algorithm. In Line 199-202 the authors report “t is necessary to make sure that the parameters of different comparative algorithms are equally set up,” I agree with that. However, is the computational effort of each algorithm the same? When we compare heuristic search methods, we need to guarantee that the same effort will be made by each algorithm, in this case, the effort will be the number o calls to the fitness function. In an interation, a different number of call of the fitness function can be made depending on the algorithm. Please clarify this point.

D.  Ant colony optimization algorithm is a probabilistic technique. Why the method IEACO was executed only 1 x? I would expect a set of execution replicas (at least 30 runs) and a mean and standard deviation analysis. Please read the following paper:
https://link.springer.com/article/10.1007/s11590-006-0011-8

Author Response

Response to Reviewer 2 Comments

Point 1:  The comparison of methods is questionable. I recommend you improve this part of the paper (for example using ideas from the following article). References: Garcia, S., Herrera, F. (2008). An extension on statistical comparisons of classifiers over multiple data sets for all pairwise comparisons. Journal of Machine Learning Research 9:2677-2694.

Response 1: Thank you for reading our manuscript carefully. The ideas in the reference is very useful to us. We have done the hypothesis test according to the reference. The hypothesis test is shown in 3.4.

Point 2:  Are the achieved results statistically significant? (between the methods)

Response 2: We have done the hypotheticals test according to the reference (Garcia, S., Herrera, F. (2008)). In the large-scale datasets, our method is statistically significant.

Point 3:  The authors define N as the number of iteration of the algorithm. In Line 199-202 the authors report “t is necessary to make sure that the parameters of different comparative algorithms are equally set up,” I agree with that. However, is the computational effort of each algorithm the same? When we compare heuristic search methods, we need to guarantee that the same effort will be made by each algorithm, in this case, the effort will be the number o calls to the fitness function. In an interation, a different number of call of the fitness function can be made depending on the algorithm. Please clarify this point.

Response 3: In the five ACOs, the number of calls of the fitness function is equal to the number of ants in an iteration. Therefore, the same number of iterations makes the fitness function called the same for each algorithm. When the same number of iterations is the same, the computational effort of each algorithm will be the same.

Point 4:  Ant colony optimization algorithm is a probabilistic technique. Why the method IEACO was executed only 1 x? I would expect a set of execution replicas (at least 30 runs) and a mean and standard deviation analysis. Please read the following paper:
https://link.springer.com/article/10.1007/s11590-006-0011-8

Response 4: Thank you for your good suggestion.

After reading the paper, we run IEACO 30 times. The mean and standard deviation analysis has been performed in 3.2. Detection Power Comparison.

Round 2

Reviewer 1 Report

Review of paper

Self-adjusting Ant Colony Optimization Based on 2 Information Entropy for Detecting Epistatic Interactions

by Boxin Guan and Yuhai Zhao

REVISED VERSION

I spent the 25th and 26th december 2018 carefully annotating the article, not only with respect to issues in syntax and grammar, but with respect to the scientific content.

I specified :

"We encourage the authors to revise their manuscript, write a cover letter with a reminder of each current defective paragraph and close below, the novel modified paragraph, for an easy check by the reviewers."

I therefore expected an answer point by point, in particular regarding the section “On the scientific / technical content” of my review. If the authors have no time to comply with these instructions, I have unfortunately no time to read the newly submitted manuscript annotated in red and showing strike-through words or sentences, to attempt to see what has changed between the first and second versions.

When the authors write:

Point 1: We encourage the authors to revise their manuscript, write a cover letter with a reminder of each current defective paragraph and close below, the novel modified paragraph, for an easy check by the reviewers.

Response 1: Thank you for reading our manuscript carefully. We have revised our manuscript by the MDPI English editing service. A manuscript of the grammar correction has been submitted.”

they show that they did not take account of my recommendations to easy my reviewer’s work. I wish to compare the first version of any modified paragraph with its second version, together with having version 2 in its entirely.

Author Response

We have taken a serious reply to each comment point to point. 

The specific responses are contained in the coverletter.

The coverletter is enclosed.
